# Mineral dust increases the habitability of terrestrial planets but confounds biomarker detection

Ian A. Boutle [1,2✉], Manoj Joshi [3], F. Hugo Lambert[1], Nathan J. Mayne [1], Duncan Lyster[1], James Manners [1,2], Robert Ridgway [1] & Krisztian Kohary[1]

Identification of habitable planets beyond our solar system is a key goal of current and future space missions. Yet habitability depends not only on the stellar irradiance, but equally on constituent parts of the planetary atmosphere. Here we show, for the first time, that radiatively active mineral dust will have a significant impact on the habitability of Earth-like exoplanets. On tidally-locked planets, dust cools the day-side and warms the night-side, significantly widening the habitable zone. Independent of orbital configuration, we suggest that airborne dust can postpone planetary water loss at the inner edge of the habitable zone, through a feedback involving decreasing ocean coverage and increased dust loading. The inclusion of dust significantly obscures key biomarker gases (e.g. ozone, methane) in simulated transmission spectra, implying an important influence on the interpretation of observations. We demonstrate that future observational and theoretical studies of terrestrial exoplanets must consider the effect of dust.

[1] College of Engineering, Mathematics and Physical Sciences, University of Exeter, Exeter EX4 4QL, UK. [2] Met Office, FitzRoy Road, Exeter EX1 3PB, UK. [3] School of Environmental Sciences, University of East Anglia, Norwich NR4 7TJ, UK. ✉email: ian.boutle@metoffice.gov.uk

Even before the discovery of the first potentially habitable terrestrial exoplanets[1], researchers have speculated on the uniqueness of life on Earth. Of particular interest are tidally locked planets, where the same side of the planet always faces the star, since this is considered the most likely configuration for habitable planets orbiting M-dwarf stars[2,3], which make up the majority of stars in our galaxy. In the absence of observational constraints, numerical models adapted from those designed to simulate our own planet have been the primary tool to understand these extraterrestrial worlds[4–8]. But most studies so far have focussed on oceanic aquaplanet scenarios, because water-rich planets are one of the likely outcomes of planetary formation models[9], the hydrological cycle is of key importance in planetary climate and the definition of habitability requires stable surface liquid water.

For a planet's climate to be stable enough for a sufficiently long period of time to allow the development of complex organisms (e.g., around 3 billion years for Earth[10]), the presence of significant land cover may be required. The carbon–silicate weathering cycle, responsible on Earth for the long-term stabilisation of $CO_2$ levels in a volcanic environment, acts far more efficiently on land than at the ocean floor[11]. Some studies have attempted to simulate the effects of the presence of land[12–16], demonstrating how it would affect the climate and atmospheric circulation of a tidally locked planet, such as Proxima b[5,7]. More specific treatments of land surface features, such as topography, have only been briefly explored[7,17].

Mineral dust is a significant component of the climate system whose effects have been hitherto neglected in climate modelling of exoplanets. Mineral dust is a class of atmospheric aerosol lifted from the planetary surface and comprising the carbon–silicate material that forms the planetary surface (it should not be conflated with other potential material suspended in a planetary atmosphere, such as condensable species (clouds) or photochemical haze). Dust is raised from any land surface that is relatively dry and free from vegetation. Dust can not only cool the surface by scattering stellar radiation, but also warm the climate system through absorbing and emitting infra-red radiation. Within our own solar system, dust is thought to be widespread in the atmosphere of Venus[18], and is known to be an extremely important component of the climate of Mars, which experiences planetary-scale dust storms lasting for weeks at a time[19,20]. Even on Earth, dust can play a significant role in regional climate[21,22] and potentially in global long-term climate[23].

Here, we demonstrate the importance of mineral dust on a planet's habitability. Given our observations of the solar system, it is reasonable to assume that any planet with a significant amount of dry, ice- and vegetation-free landcover, is likely to have significant quantities of airborne dust. Here, we show for the first time that mineral dust plays a significant role in climate and habitability, even on planets with relatively low land fraction, and especially on tidally locked planets. We also show that airborne dust affects near-infra-red transmission spectra of exoplanets, and could confound future detection of key biomarker gases such as ozone and methane. Airborne mineral dust must therefore be considered when studying terrestrial exoplanets.

## Results

### Schematic mechanisms

We consider two template planets, a tidally locked planet orbiting an M dwarf (denoted TL), with orbital and planetary parameters taken from Proxima b, and a non-tidally locked planet orbiting a G dwarf (denoted nTL), with orbital and planetary parameters taken from Earth. The choice of parameters is merely to give relatable examples; the results presented are generic and applicable to any planet in a similar state.

We also consider the planets to be Earth-like in atmospheric composition, i.e., 1-bar surface pressure and a nitrogen-dominated atmosphere, as this is the most well-understood planetary atmosphere, and the only one known to be inhabited. For each of these planets, we consider a range of surface land-cover amounts and configurations, designed to both explore the parameter space that may exist and understand in which scenarios dust is important. Starting from well-understood aquaplanet simulations[6] derived using a state-of-the-art climate model[24], we increase the fraction of land in each model grid cell equally, until the surface is completely land. This experiment (denoted Tiled) acts to both increase the amount of land available for dust uplift, whilst reducing the availability of water, and thus the strength of the hydrological cycle, without requiring knowledge of continent placement. For the TL case, we additionally conduct simulations in which a continent of increasing size is placed at the substellar point (denoted Continents). This produces a fundamentally different heating structure from the central star[15], and significantly increases the effect of the dust for small land fractions, whilst allowing a strong hydrological cycle to persist.

For each planet and climate configuration, we run two simulations, one without dust, called NoDust, equivalent to all previous studies of rocky exoplanets, and one in which dust can be lifted from the land surface, transported throughout the atmosphere and interacted with the stellar and infra-rad radiation and atmospheric water, called Dust.

The mechanisms through which dust affects planetary climate are illustrated in Fig. 1. Incoming stellar radiation is concentrated over a smaller area on the TL planet (Fig. 1a) compared with the nTL case (Fig. 1b). Strong surface winds on the dayside of TL allow for much greater uplift of dust than the equatorial doldrums of nTL. The super-rotating jet on TL is more efficient at transporting this dust to cooler regions on the nightside (Fig. 1c), than the more complex atmospheric circulation on nTL is at transporting dust to the poles (Fig. 1d). The radiative forcing, or change in surface energy balance caused by airborne dust, is therefore weaker for nTL than TL. As a result, the nTL planet is broadly cooled by dust (Fig. 1j) because the airborne dust's infra-red greenhouse effect (Fig. 1h) is cancelled out by the stellar radiation changes due to scattering and absorption by airborne dust (Fig. 1f). However, the TL planet is strongly cooled on its warm dayside by similar mechanisms, but warmed on its nightside (Fig. 1i) because the airborne dust's infra-red greenhouse effect (Fig. 1g) has no stellar radiation change to offset it (Fig. 1e).

### Habitable zone changes

Figure 2 shows two key metrics we use to quantify the outer and inner edges of the habitable zone for our template planets. The outer edge of the habitable zone is likely to be controlled by the temperature at which $CO_2$ condenses[25], which for the concentrations and surface pressures considered here, is at $\approx$125 K. Keeping the minimum temperature above this threshold is therefore a key requirement for maintaining a $CO_2$ greenhouse effect, and preventing a planet's remaining atmospheric constituents from condensing out. Figure 2a shows that for the TL case, the presence of dust always acts to increase the minimum temperature found on the planet (blue and magenta lines). The effect of dust is to sustain a greenhouse effect at a lower stellar irradiance than when dust is absent, implying that dust moves the outer edge of the habitable zone away from a parent star. The effect is not especially sensitive to the specific arrangement of the land (magenta vs. blue lines in Fig. 2a), but is very sensitive to the fraction of the surface covered by land; the approximate change in stellar radiation at the outer edge of the habitable zone is over 150 W m$^{-2}$ for a totally land-covered

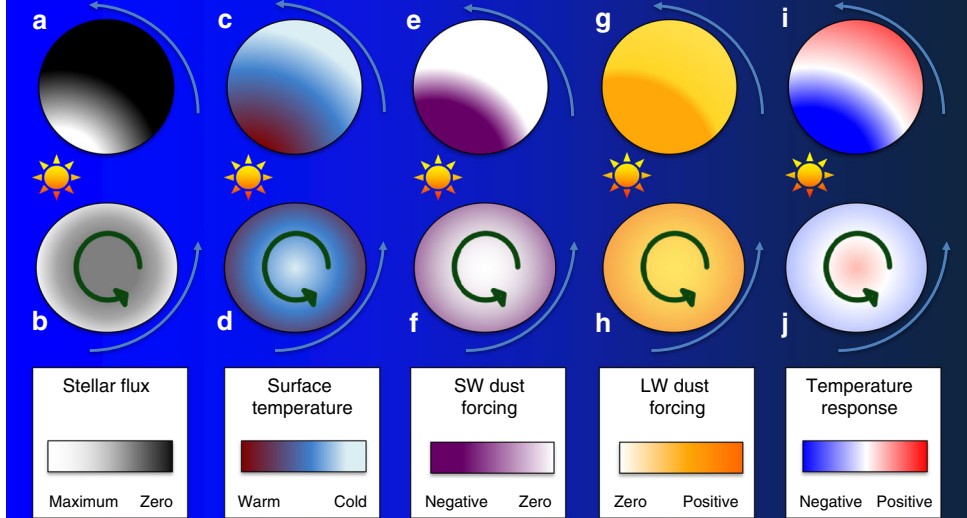

**Fig. 1 Schematic showing the effect dust has on the climate of planets.** For a tidally locked planet (**a**) and non-tidally locked planet (**b**), panels **a**–**d** show the base state of the planets, **e**–**h** show the short- (stellar) and long-wave (infra-red) forcing (change in surface energy balance) introduced by dust and **i**–**j** show the resultant effect of the forcing on the surface temperature. Blue arrows show the motion of the planet around the star, and green arrows show the rotation of the planet relative to the star.

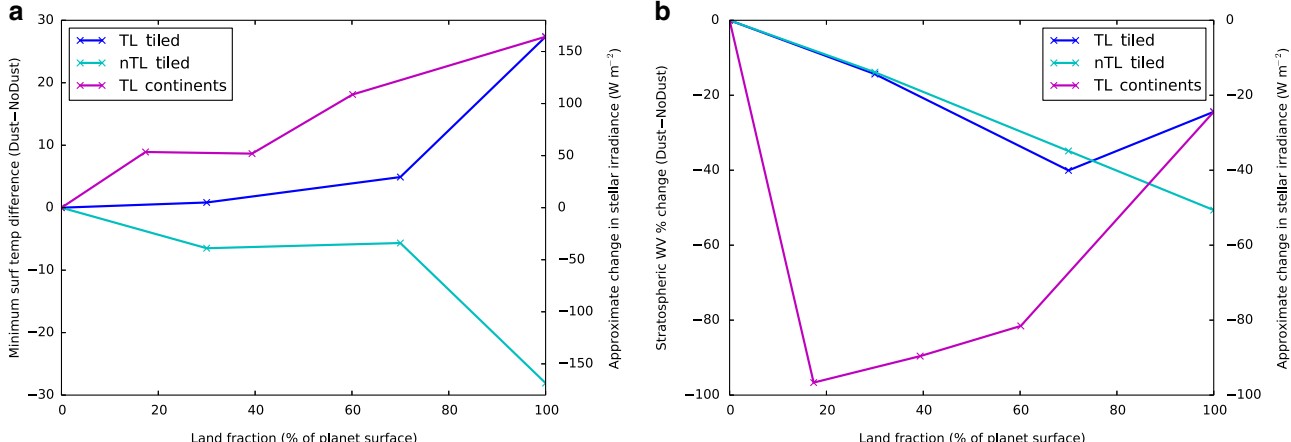

**Fig. 2 Effect of dust on habitable zone boundary indicators.** Differences in **a** minimum surface temperature, and **b** stratospheric (≈50 hPa) water-vapour content, between simulations with (Dust) and without (NoDust) mineral dust, as a function of land fraction. The different planetary and surface setups are shown in the legend. Approximate equivalent changes in stellar irradiance required to achieve similar responses in a dust-free planet are shown on the right axis.

planet, but even up to 50 W m$^{-2}$ for a planet with the same land coverage as Earth. Such results are in stark contrast to the nTL case, for which dust always acts to reduce the minimum surface temperature (cyan line in Fig. 2a), moving the outer edge of the habitable zone inwards.

The inner edge of the habitable zone is likely to be controlled by the rate at which water vapour is lost to space, often termed the moist greenhouse[26–28]. The strength of the water-vapour greenhouse effect increases with surface temperature, eventually leading to humidities in the middle atmosphere that are large enough to allow significant loss of water to space. Stratospheric water-vapour content is therefore a key indicator of when an atmosphere will enter a moist greenhouse. Figure 2b shows that for all our simulations, the effect of dust is to reduce stratospheric water-vapour content, i.e., dust suppresses the point at which a moist greenhouse will occur and moves the inner edge of the habitable zone nearer to the parent star. The effect on the habitable zone can be approximately quantified by utilising additional simulations done with increased or reduced stellar flux

and a constant tiled land fraction of 70% (Table 3). They show that stratospheric water vapour scales approximately logarithmically with stellar flux, allowing us to infer that the 30–60% reduction in stratospheric water vapour caused by dust (shown in Fig. 2b) roughly corresponds to a stellar flux reduction of 30 – 60 W m$^{-2}$. In contrast to the effect on the outer edge, both our TL and nTL simulations result in a reduction in stratospheric water vapour when including dust, demonstrating that the inward movement of the inner edge of the habitable zone is a ubiquitous feature of atmospheric dust. However, here the magnitude of the effect is more dependent on the arrangement of the land, and therefore more uncertain. Supplementary Notes 1 and 2 give more details on this.

In summary, radiatively active atmospheric dust increases the size of the habitable zone for our tidally locked planets, both by moving the inner edge inwards and outer edge outwards. For our non-tidally locked planets, both the inner and outer edges of the habitable zone move inwards, so the consequences for habitable zone size depend on which effect is stronger. The exact size of the

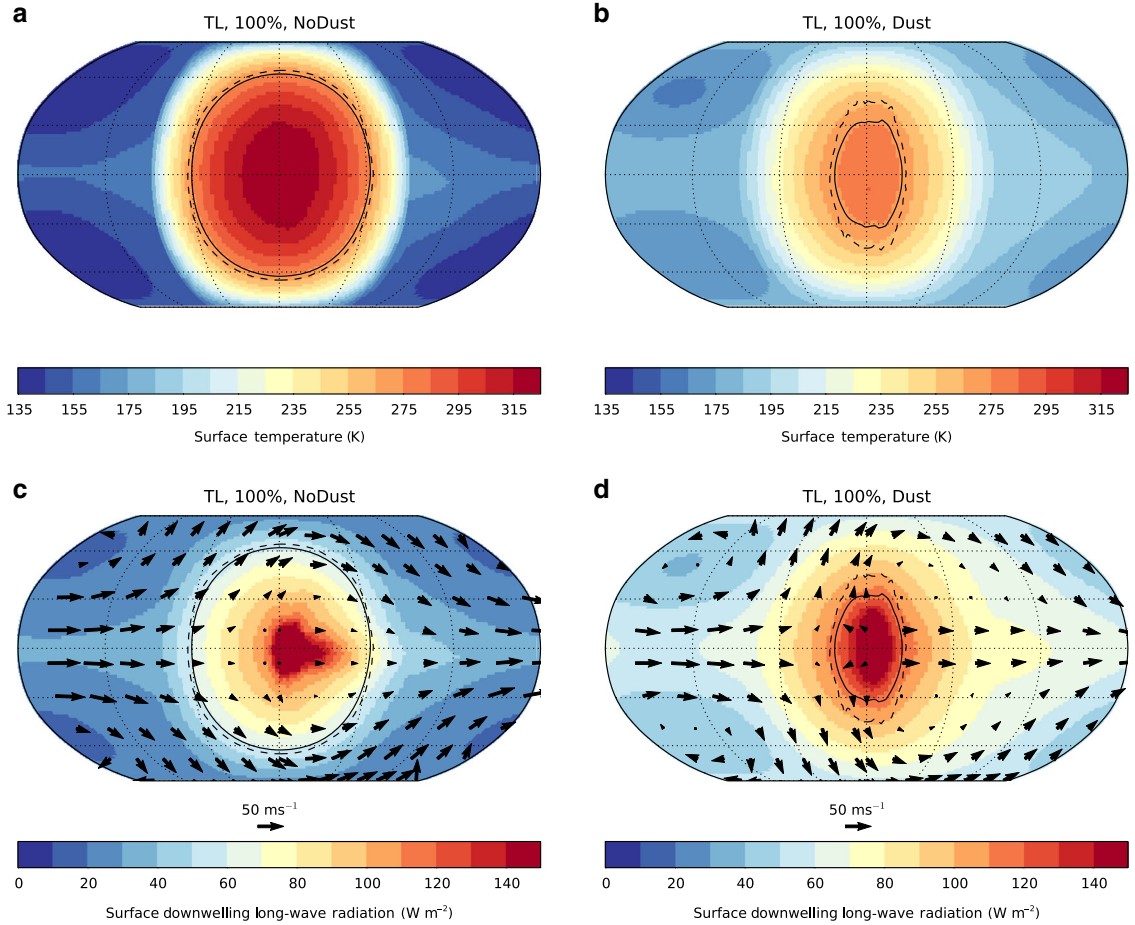

**Fig. 3 Mechanisms driving surface temperature change.** Surface temperature (**a**, **b**) and surface downwelling long-wave radiation (**c**, **d**), from the TL case with 100% landcover for the NoDust (**a**, **c**) and Dust (**b**, **d**) simulations. Also shown are the mean (solid) and maximum (dashed) 273-K contours, and wind vectors at 8.5 km (≈300 hPa, in **c** and **d**).

habitable zone is a subject of much debate[2,27–29], and how well our results can be extrapolated to previous estimates of its size are covered in the 'Discussion'. But to illustrate the potential importance of dust, conservative estimates from Kasting et al.[2] suggest a stellar irradiance range of ~750 W m$^{-2}$ from the inner to the outer edge. Figure 2a shows that the effect of dust is equivalent to changing the stellar irradiance by up to 150 W m$^{-2}$, thereby moving the outer edge of the habitable zone by up to 10% in either direction.

Figure 3 illustrates the effects of dust on climate for the TL case in more detail. We show the results for the 100% land simulation where the dust effect is the strongest, and although the effect is weakened with lower fractions of land, the mechanisms remain the same. The dust particles are lifted from the surface on the dayside of the planet, since uplift can only occur from non-frozen surfaces. There they are also strongly heated by incoming stellar radiation. The larger particles cannot be transported far before sedimentation brings them back to the surface, but the smaller particles can be transported around the planet by the strong super-rotating jet expected in the atmospheres of tidally locked planets[30]. The smaller dust sizes are therefore reasonably well-mixed throughout the atmosphere, and able to play a major role in determining the radiative balance of the nightside of the planet. This highlights an important uncertainty not yet discussed—our assumption that surface dust is uniformly distributed amongst all size categories. As only the small- to mid-sized dust categories play a major role in determining the planetary climate, increasing or decreasing the amount of surface dust in these categories can

increase or decrease the quantitative effects. Similarly, the precise formulation of the dust-uplift parameterisation can have a similar quantitative effect on the results presented. More discussion is given in Supplementary Note 1, but neither uncertainty changes the qualitative results presented.

The coldest temperatures are found in the cold-trap vortices on the nightside of the planet (Fig. 3a), which without dust are ≈135 K. The effect of dust is to raise the temperature reasonably uniformly by ≈25 K across the nightside of the planet (Fig. 3b), significantly raising the temperature of the cold traps above the threshold for $CO_2$ condensation. The increase in surface temperature arises because of a corresponding increase in the downwelling infra-red radiation received by the surface of the nightside of the planet (Fig. 3c, d), which is approximately doubled compared with a dust-free case.

Supplementary Note 2 explores some of the more detailed responses of the different land surface configurations that are shown in Fig. 2. However, in all cases, the effect on the habitable zone and mechanisms for the change are consistent with those described above, and are likely to be significant for any continental configuration and even for low fractions of land.

**Simulated observations.** A key question regarding airborne mineral dust is how it would affect the interpretation of potential future spectra of terrestrial exoplanets. Figure 4 presents synthetic observations created from our model output combined with the PandExo simulator[31] of the NIRSpec (G140M, G235M and G395M modes) instrument on the James Webb Space Telescope

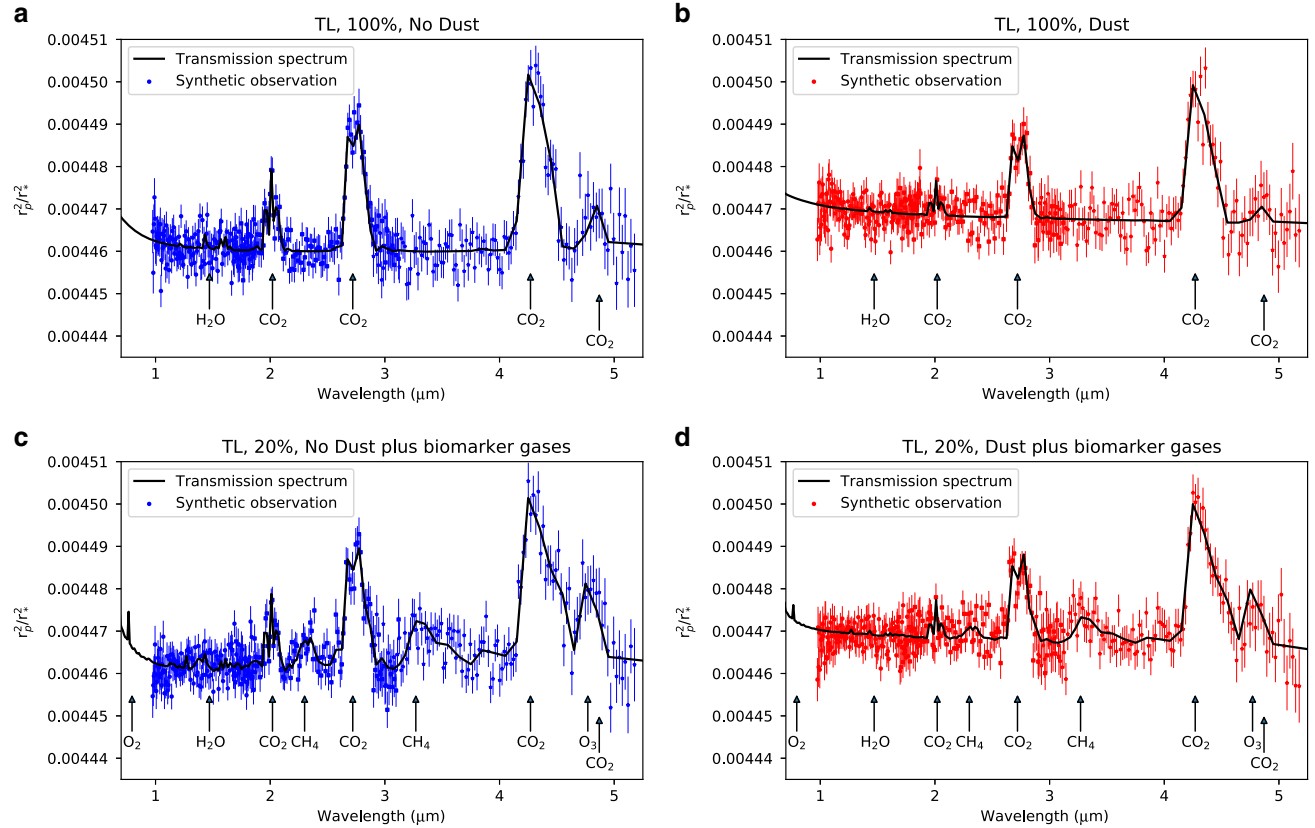

**Fig. 4 Effect of dust on planetary observations.** Simulated transmission spectra (black) and synthetic JWST observations (blue/red), from 15 transits for a dusty (**b**, **d**) and non-dusty (**a**, **c**) tidally locked planet, orbiting an M dwarf of apparent magnitude similar to Proxima Centauri, with 100% land coverage and no potential biomarker gases (**a**, **b**), and 20% landcover and biomarker gases (**c**, **d**). A one-standard deviation error on the synthetic observation is shown.

(JWST), following the method described in Lines et al.[32]. We focus here on the TL case, and compare the relatively dry 100% land-cover simulation with the 20% landcover arranged as a continent simulation, to demonstrate how even planets with low dust loading and a strong hydrological cycle can be affected. We additionally consider the 20% land-cover simulation to have an atmospheric composition that is Earth-like, i.e., it contains the key observable potential biomarker gases oxygen, ozone and methane[33] in present-day Earth concentrations. Adding these gases does not greatly affect the climatic state[6], but can significantly alter the observed spectra. We consider a target object with the apparent magnitude of Proxima Centauri, as stars near this range are the most likely candidates for observing in the near future (Proxima b itself does not transit[34], but that does not invalidate our results for similar planets around similar stars). We discuss how our results change for dimmer stars such as TRAPPIST-1 in Supplementary Note 3.

Figure 4 shows that airborne dust effectively introduces a new continuum absorption into the spectrum, which completely obscures many of the minor absorption peaks similar to previous studies of hotter planets[32,35], some of which are associated with potential biomarker gases, such as methane (2.3 and 3.3 μm) and ozone (4.7 μm). An oxygen feature at 0.76 μm is also significantly obscured in the dusty spectrum, and although it falls outside the spectral range of JWST, is similarly unlikely to be prominent enough if it was within the observable spectrum. Importantly, biomarker gas features are obscured even when dust loading is relatively low (Fig. 4c, d), i.e., even relatively wet planets with a strong hydrological cycle are prone to having important spectral peaks being obscured from observation by dust.

## Discussion

Given the radiative properties of dust, and the dependence of its impact on the climate on land fraction (Fig. 2), it could potentially produce a strong negative feedback for planets undergoing significant water loss at the inner edge of the habitable zone. As water is lost and the fraction of the surface covered by ocean decreases, the amount of dust that is suspended in the atmosphere will likely increase, which in turn cools surface temperatures, quite dramatically in the case of a tidally locked planet, reducing the amount of water vapour in both the lower and middle atmosphere. Airborne dust can therefore act as a temporary brake on water loss from planets at the inner edge of the habitable zone in a similar manner to the ocean fraction/water-vapour feedback[13]. However, how dust interacts with other mechanisms affecting the inner edge of the habitable zone requires further study. For example, the potential bistable state of planets with water locked on the nightside[36], which may also widen the habitable zone, may be partly offset by the presence of dust if the warmer nightside (due to the mechanisms discussed here) allows some of the water to be liberated back to the dayside.

Estimates of the outer edge of the habitable zone[29] are also typically made with much higher $CO_2$ partial pressures than those considered here (up to 10 bar). It is unclear that such high $CO_2$ concentrations could be achieved in the presence of land, due to increased weathering activity preventing further $CO_2$ buildup[37]. If they could, the quantitative effect of dust will depend on a range of compensating uncertainties. For example, dust uplift should be enhanced due to higher surface stresses in a higher-pressure atmosphere. However, dust transport to the nightside may be

reduced in the weaker super-rotating jet due to reduced day–night-temperature contrasts[38].

Our results have implications for studies of the history of our own planet before terrestrial vegetation covered large areas, with a particular example being the faint young Sun problem of Archaean Earth[39]. The land masses that are believed to have emerged during this period will have been unvegetated, and therefore a significant source of dust uplift into the atmosphere if dry and not covered in ice. As we have shown, this dust would have a cooling effect on the planetary climate, potentially making the faint young Sun problem harder to resolve. However, it is also possible that microbial mats might have covered large areas of the land surface before vegetation evolved. The exact nature of such cover, and how much it would hinder dust lifting into the atmosphere, is yet to be quantified.

It is clear that the possible presence of atmospheric dust must be considered when interpreting observations. The feature-rich spectrum observed from a dust-free atmosphere containing water vapour, oxygen, ozone and methane (Fig. 4c) is transformed into a flat, bland spectrum where only major $CO_2$ peaks are visible above the background dust continuum (Fig. 4d). Observations returning a spectrum such as this could easily be misinterpreted as being caused by a dry atmosphere containing only nitrogen and $CO_2$, i.e., Fig. 4d interpreted as Fig. 4a. The result would be a potentially very interesting planet being characterised as dry, rocky and lifeless. On the other hand, if spectra are obtained that can unambiguously place a limit on dust generation, such results imply a mechanism that inhibits dust lifting, whether it be some combination of a very small land fraction, significant ice or vegetation cover or other dust-inhibiting mechanisms: such a result would also be of great interest to those interpreting observations.

Finally, our results have wide-ranging consequences for future studies of the habitability of terrestrial rocky planets. Such studies should include models of airborne dust as well as observational constraints. Furthermore, our results strongly support the continued collaboration between observational and modelling communities, as they demonstrate that observations alone cannot determine the size of the habitable zone: it crucially depends on properties of the planetary atmosphere, which are presently only accessible via climate modelling.

## Methods

**Numerical model set-up.** Our general circulation model of choice is the GA7 science configuration of the Met Office Unified Model[24], a state-of-the-art climate model that incorporates within it a mineral dust parameterisation[40,41], which includes uplift from the surface, transport by atmospheric winds, sedimentation, and interaction with radiation, clouds and precipitation. The parameterisation comprises nine bins of different-sized dust particles (0.03–1000 μm). The largest three categories (>30 μm) represent the precursor species for atmospheric dust; these are the large particles that are not electrostatically bound to the surface, but can be temporarily lifted from the surface by turbulent motions. They quickly return to the surface under gravitational effects, and as such are not transported through the atmosphere (they do not travel more than a few metres). However, they are important because their subsequent impact with the surface is what releases the smaller particles into the atmosphere. These smaller six categories (<30 μm) are transported by the model's turbulence parameterisation[42], moist convection scheme[43] and resolved atmospheric dynamics[44]. They can return to the surface under gravitational settling, turbulent mixing and washout from the convective or large-scale precipitation schemes[45]. The absorption and scattering of short- and long-wave radiation by dust particles are based on optical properties calculated from Mie theory, assuming spherical particles, and each size division is treated independently.

The land surface configuration is almost identical to that presented in Lewis et al.[15], i.e., a bare-soil configuration of the JULES land surface model set to give the planet properties of a sandy surface. Our key difference is the use of a lower-surface albedo (0.3). The land is at sea-level altitude with zero orography and a roughness length of $1 \times 10^{-3}$ m for momentum and $2 \times 10^{-5}$ m for heat and moisture (although these are reduced when snow is present on the ground). The soil moisture is initially set to its saturated value, but evolves freely to its equilibrium state. Land is assumed to comprise dust of all sizes, uniformly distributed across the range. The dust parameterisation is used in its default Earth set-up, and naturally adapts to the absence of vegetation, suppresses uplift in wetter regions and prevents it from frozen or snow-covered surfaces. The ocean parameterisation is a slab ocean of 2.4-m mixed-layer depth with no horizontal heat transport, as was used in Boutle et al.[6], and includes the effect of sea ice on surface albedo following the parameterisation described in Lewis et al.[15]. It is worth noting that whilst the set-up implies an infinite reservoir of both water in the ocean and dust on the land, this is not actually a requirement of the results—all that is required is enough water/dust to support that which is suspended in the atmosphere and deposited in areas unfavourable for uplift (e.g., the nightside of the TL planet), and that some equilibrium state is achieved whereby additional water/dust deposited in areas unfavourable for uplift can be returned to areas where uplift can occur, e.g., basal melting of glaciers.

The orbital and planetary parameters for our two template planets are given in Table 1. To simplify the analysis slightly, both planets are assumed to have zero obliquity and eccentricity. Atmospheric parameters are given in Table 2. Again, for simplicity, we assume that the atmospheric composition is nitrogen-dominated with trace amounts of $CO_2$ for the control experiments investigating the role of dust on the atmosphere. However, because of the important role that potential biomarker gases, such as oxygen, ozone and methane, have in the transmission

### Table 1 Orbital and planetary properties.

| | TL | nTL |
|---|---|---|
| Semi-major axis (AU) | 0.0485 | 1.00 |
| Stellar irradiance (W m$^{-2}$) | 881.7 | 1361.0 |
| Stellar spectrum | Proxima Centauri | The Sun |
| Orbital period (Earth days) | 11.186 | 365.24 |
| Rotation rate (rad s$^{-1}$) | $6.501 \times 10^{-6}$ | $7.292 \times 10^{-5}$ |
| Eccentricity | 0 | 0 |
| Obliquity | 0 | 0 |
| Radius (km) | 7160 | 6371 |
| Gravitational acceleration (m s$^{-1}$) | 10.9 | 9.81 |

Shown for the tidally locked (TL) and non-tidally locked (nTL) simulations.

### Table 2 Atmospheric parameters used in this study.

| Parameter | Control simulations | Synthetic observations |
|---|---|---|
| Mean surface pressure (Pa) | $10^5$ | $10^5$ |
| $R$ (J kg$^{-1}$ K$^{-1}$) | 297 | 287.05 |
| $c_p$ (J kg$^{-1}$ K$^{-1}$) | 1039 | 1005 |
| $CO_2$ MMR (kg kg$^{-1}$) / ppm | $5.941 \times 10^{-4}$ / 378 | $5.941 \times 10^{-4}$ / 391 |
| $O_2$ MMR (kg kg$^{-1}$) / ppm | 0 | 0.2314 / $209 \times 10^3$ |
| $O_3$ MMR (kg kg$^{-1}$) / ppm | 0 | $2.4 \times 10^{-8}$ / 0.015 (min) |
| | | $1.6 \times 10^{-5}$ / 9.66 (max) |
| $CH_4$ MMR (kg kg$^{-1}$) / ppm | 0 | $1.0 \times 10^{-7}$ / 0.18 |
| $N_2O$ MMR (kg kg$^{-1}$) / ppm | 0 | $4.9 \times 10^{-7}$ / 0.32 |

Shown for the baseline simulations with a simple nitrogen plus trace $CO_2$ atmosphere, and the synthetic observations with a more Earth-like atmosphere. Gas quantities are given in mass-mixing ratio (MMR) and parts per million (ppm).

**Table 3 Full list of the 28 experiments presented in this study.**

| Land fraction | TL-continents | TL-tiled | nTL-tiled |
|---|---|---|---|
| 0 | Control | Control | Control |
| 20 | Control; synthetic | | |
| 30 | | Control | Control |
| 40 | Control | | |
| 60 | Control | | |
| 70 | | Control; SI − 244; SI + 394 | Control; SI − 161; SI + 139 |
| 100 | Control; k1 = 2; k1 = 2, small | | Control |

The orbital and planetary parameters (TL and nTL) are taken from Table 1, the atmospheric parameters (Control, Synthetic) are taken from Table 2 and Continents/Tiled is explained in the text (N.B. for 0 and 100% land fraction, the Continent or Tiled set-up is identical). Additional experiments denoted SI contain changes in the stellar irradiance from those quoted in Table 1, and k1 varies the dust uplift as discussed in Supplementary Note 1. Most experiments contain two variants—with and without radiatively interactive mineral dust.

spectrum, when discussing simulated observables, we include these gases at an abundance similar to that of present-day Earth.

Table 3 summarises the 28 experiments and their parameters that are described in this paper.

## Data availability
All data used in this study are available from https://doi.org/10.24378/exe.2284.

## Code availability
The Met Office Unified Model is available for use under licence, see http://www.metoffice.gov.uk/research/modelling-systems/unified-model.

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

## Acknowledgements
I.B. and J.M. acknowledge the support of a Met Office Academic Partnership secondment. We acknowledge use of the MONSooN system, a collaborative facility supplied under the Joint Weather and Climate Research Programme, a strategic partnership

between the Met Office and the Natural Environment Research Council. NM was partly funded by a Leverhulme Trust Research Project Grant that supported some of this work alongside a Science and Technology Facilities Council Consolidated Grant (ST/R000395/1). This work also benefited from the 2018 Exoplanet Summer Programme in the Other Worlds Laboratory (OWL) at the University of California, Santa Cruz, a programme funded by the Heising-Simons Foundation.

## Author contributions

I.B. ran the simulations and produced most of the figures and text. M.J. had the original idea and provided guidance, Fig. 1 and contributions to the text. F.H.L. and N.M. provided guidance and contributions to the text. D.L. investigated the role of continents as a Masters project. J.M. provided scientific and technical advice. R.R. produced the synthetic observations in Fig. 4. K.K. provided technical support.

## Competing interests

The authors declare no competing interests.
