## [Peer Review File · Nature Communications]

Reviewers' comments:

Reviewer #1 (Remarks to the Author):

This paper examines the climatic effect of dust loading into an atmosphere in two different scenarios. One with an Earth like world with increasing amounts of land to sea while keeping Earth's modern rotation rate. In the other case they look at a tidally locked Proxima Centauri b type planet with increasing amounts of land at the substellar point and "tiled."

This is a unique approach I've not seen before in the literature and has serious implications for observing terrestrial planets with JWST and other future instruments. I recommend publication, but I would like the authors to make a few changes. I don't need to see the manuscript again if they agree to these modest changes. Of course if other referees demand substantial revisions then it may be necessary to see it again afterward.

1.) It is never discussed what type of ocean is used. Is it a simple mixed layer depth X ocean w/o horizontal heat transport? Is it a fully-coupled bathtub ocean? What is it?! Ocean dynamics are important in climate studies, and we should know what it being used.

2.) Line 8: it is not until line 109 that the term "radiatively active mineral dust" is mentioned. I think that belongs here in the Abstract.

3.) Lines 23-25 "But most studies so far have focussed on oceanic\aquaplanet" scenarios, because of the importance of the hydrological cycle and the definition of habitability requiring stable surface liquid water."

Yet there have been a few large parameter studies that have used non-aquaplanet scenarios, in particular: Yang et al. 2014 ApJL 787, L2, and Way et al. 2018 ApJS 239, 24. You might mention them and/or others.

4.) Lines 31-32 "More specific treatments of land surface features such as topography remain unexplored."

This is a rather strong statement. It has been explored in a few areas, just not extensively. Your reference 7 (Del Genio et al.) does have Prox b simulations with an aquaplanet set up AND Earth topography (See their Table 4 and in particular #18 Day-Land) At least one more I am aware of is Way et al. 2016 GRL 43, 8376 where they explicitly compare a Venus topography to an Earth topography each with an ocean of similar depth (see Simulation A versus C in Table 1). Although there is no discussion of dust in either of these works.

5.) I really like your Figure 1!

6.) From Figure 2 it looks to me like you did 4 simulations "TL tiled", 4 "nTL Tiled" and 5 "TL Continents." It would be nice to have a table of all of the simulations with their respective parameters listed. If you don't have room in the main text perhaps in the Suppl Mat?

7.) Lines 100-103: Sorry, but I don't quite understand this statement. First aren't you extrapolating to get the 70% simulation for "TL Continents?" Please say so if that's the case. Then you say 20-50% reduction in 2b. But if you mean only at 70% land fraction I don't see the same range. Could you be a bit more explicit? The same

applies for the comment about 20-50 W/m².

8.) Figure 3 caption: 8.3 km. Can you tell the reader what this is in pressure? hPa or mb would be fine.

9.) Figure 4 b & d. Could you please add to the title of each 'plus biomarker gases' or something like that to make it easier for the reader to distinguish w/o having to read your caption so carefully?

10.) Reference 24: the last author's name is missing a 'c.' Stamenkovic

11.) In your methods section regarding your dust parameterization. Nowhere is it explicitly mentioned whether you are using the default Earth parameterization for your dust, or if you used your model in interactive mode with your dust sizes to generate a proper parameterization w/o Earth vegetation included. Please tell the reader.

12.) Table 2: Could you possibly add a column that lists the partial pressures for CO₂, O₂, O₃, CH₄ and N₂O in ppmv? It's easier to compare with other work w/o the reader having to do the work themselves. Also nowhere in the text does the reader learn what "MMR" means.

13.) Line 282: Could you please be a bit more explicit about "The surface is essentially flat" so that the reader doesn't have to read Lewis et al. and hope it's the same? Some mention of roughness length would also seem appropriate since it will affect dust lifting.

On to the Supplementary Material:

1.) Line 67 add TL: "... between TL land surface configurations..."

2.) Figure S4 certainly makes for depressing, but useful reading. Although it was only recently published it would be nice if you mentioned it in the context of Figure 5 from Fauchez et al. 2019 ApJ 887, 194, even if theirs is for a noise free spectrum, they do mention in their Table 4 how many transits you need for a given S/N for a specific feature. Between your dust and their clouds and hazes, the prospects look bleak for detecting anything other than CO₂!

Reviewer #2 (Remarks to the Author):

Overall, I think that the paper is well written, well organized and deals with a topic that has been largely overlooked in the literature. I am sure the results of this paper will be of interest to the community, as it brings first insights on the effects (and underlying mechanisms) that dust may have on the climate of temperate, terrestrial exoplanets.

However, there are two important points I would like the authors to clarify/correct/expand in their manuscript before I recommend it for publication.

My first main concern is about the numerical parameterization of the dust cycle. The authors mentioned in their manuscript that they used the mineral dust parameterization of Woodward et al. 2011 (technical report). The dust emission scheme seems to be tuned and designed with empirical coefficients to reproduce the Earth dust cycle, and I am concerned whether it can be

applied as is on other planets.

It is well known in the planet Mars literature that the dust cycle (and in particular the dust injection) is difficult to model (Newman et al. 2002, Kahre et al. 2006, Mulholland et al. 2015, Bertrand et al. 2019) with 3-D GCMs because it depends on subtle sub-grid scale parameterizations.

Can the authors reproduce roughly the dust cycle of Mars with their model? (any reference of previous work done with the UK Met Office GCM? Any additional run?)

Can the authors explore how sensitive are their results to their dust emission parameterization? (e.g. by doing additional runs with various wind threshold friction velocities?)

Last, the authors assumed that the reservoir of dust is infinite. Is it reasonable especially for planets with low land fraction? A discussion on sources versus sinks would be welcome.

My second main concern is about the interpretation and implication of their results regarding the habitability of terrestrial exoplanets (see also my more detailed comments hereafter).

Although I am convinced by the results obtained by the authors for 1 bar N₂-dominated Earth-like planet, I am not sure that they extrapolate well – unlike what is stated by they authors, even in the title of this manuscript – to the habitability of planets in general.

The authors performed indeed 3D GCM experiments for tidally locked planets with a 1 bar Earth-like atmosphere for which they showed that the dust cycle tend to stabilize the climate. This stems from the fact that dust reflects efficiently light on the dayside of the planet, thus cooling the dayside temperature of irradiated planets which reduces the stratospheric water vapour content and delay the moist greenhouse ; but warms the nightside, thus preventing from CO₂ surface condensation, thus stabilizing the climate of weakly irradiated planets.

(1) On relatively dry (and thus dusty) planets, you would expect the water to migrate on the nightside of the planet. See Abe et al. 2011, Menou 2013, Leconte et al. 2013, Yang et al. 2014, Turbet et al. 2016, Kodama et al. 2018, 2019. From that perspective, the scenario TL continents is relevant. You show for a 1bar N₂-dominated Earth-like atmosphere that if you have a dusty atmosphere, you expect to decrease the stratospheric water vapour content (Fig 2b, red curve). But you also expect to warm the nightside of the planet (Fig 2a, red curve), which should boost the hydrological cycle on the nightside of the planet. This means more evaporation, but also a much more efficient water flow on the surface. This can be seen in your Figure 3a and 3b where adding dust increase the surface temperature of the two nightside gyres which would drastically increase water ice flow from nightside to dayside. This should increase in turn the area of wet regions, which is the key ingredient for determining the position of the inner edge of the HZ according to Abe 2011, Kodama et al. 2018, 2019.

In summary, dust here would decrease the habitability of terrestrial planets.

(2) As the CO₂ partial pressure increases (e.g. up to several bars) in a terrestrial planet, the heat redistribution between day and nightside also increases (see for example Turbet et al. 2018 Fig. 11, Fauchez et al. 2019 Fig. 10). This should partly move a planet toward a 'nTL'-like configuration where the effect of dust on the nightside should be significantly lowered (see your Figure 2a, green curve).

For thick CO₂-dominated planets representative of planetary atmospheres used to estimate the position of the outer edge of the Habitable Zone (Kopparapu et al. 2013, Kadoya & Tajika 2019 and many other references), your Figure 2a seems to predict the opposite effect of your claim. In summary, dust here would again decrease the habitability of terrestrial planets.

To end on that second main comment, new additional runs (e.g. taking self-consistently into

account the land fraction as a function of the temperature of the nightside ; e.g. with thick multibars CO₂-dominated atmospheres) would be needed to support the claim that dust do really increase the habitability of terrestrial planets as stated in the title, in the abstract and throughout the text.

Last but not least, I have appended below a detailed list of remarks which mainly consists of minor remarks and additional clarifications on my two main concerns.

-l18 - « the first potentially-habitable terrestrial exoplanets1 »

The reference points to Wordsworth et al. (2011) which studied GJ581d. However, the planet was shown later not to exist. It was likely an artefact of stellar activity. See Robertson et al. 2014 (<https://ui.adsabs.harvard.edu/abs/2014Sci...345..440R/abstract>). It may be more relevant to cite the first non-disputed potentially habitable terrestrial-size planet : Kepler-62f. Ref : Borucki et al. 2013 (<https://ui.adsabs.harvard.edu/abs/2013Sci...340..587B/abstract>)

-l21 – ref 2 and 3

It may be more relevant to cite :

Dole 1964 (<https://ui.adsabs.harvard.edu/abs/1964hpfm.book.....D/citations>)
Kasting et al. 1993 (<https://ui.adsabs.harvard.edu/abs/1993Icar..101..108K/abstract>)
Barnes 2017 (<https://ui.adsabs.harvard.edu/abs/2017CeMDA.129..509B/abstract>)

-l23-25 - « But most studies so far have focussed on oceanic "aquaplanet" scenarios, because of the importance of the hydrological cycle and the definition of habitability requiring stable surface liquid water. »

This is also because water-rich planets are one of the likely outcome of planetary formation models. See for instance Tian & Ida 2015 (<https://ui.adsabs.harvard.edu/abs/2015NatGe...8..177T/abstract>).

-l26-28 - « For a planet's climate to be stable enough for a sufficiently long period of time to allow the development of complex organisms (e.g. around 3 billion years for Earth 9), the presence of significant land cover is likely required »The surface is essentially fla

This is extremely speculative! I suggest to rephrase « is likely required » by « may be required ».

- l28 - « responsible for the stabilisation of CO₂ levels »

Please add « responsible on Earth for the long-term stabilisation of CO₂ levels »

- l30 - « to simulate the effects of the presence of land 11,12,13 »

You may also cite the previously mentioned Turet et al. 2016 (land planet) and Del Genio (ocean with continents) et al. 2019.

- l40 - « the presence of dust in the atmosphere of super-Earth planets has been reported 17 »

The reference 17 looks weird to me. The flat spectrum reported in Kreidberg et al. 2014 could be as well due to the presence of non-mineral clouds or photochemical hazes. Please give additional

evidence/reference proving this must be dust.

- I85-86 - « The effect of dust is to sustain a CO₂ greenhouse effect at a lower stellar irradiance than when dust is absent »

Do you mean « to sustain a greenhouse effect »? Is this a typo?

- I86-87 : « implying that dust moves the outer edge of the habitable zone away from a parent star »

I don't agree with that. As the CO₂ partial pressure increases, the heat redistribution between day and nightside also increases. This can be observed in Turbet et al. 2018 (<https://ui.adsabs.harvard.edu/abs/2018A%26A...612A..86T/abstract>) Figure 11, Fauchez et al. 2019 (<https://ui.adsabs.harvard.edu/abs/2019ApJ...887..194F/abstract>) Figure 10, and many other references.

So I guess this should partly move the planet toward a 'nTL'-like configuration where the effect of dust on the nightside should be significantly lowered (see your Figure 2a, green curve).

This could be tested by running your GCM with an atmosphere of e.g. 10bar dominated by CO₂.

Also by looking at the Figure 17 of Forget et al. 2013 (<https://ui.adsabs.harvard.edu/abs/2013Icar..222...81F/abstract>) that did similar simulations for the case of early Mars, it seems the effect of dust is relatively low. Could you comment on that?

- Figure 2

I don't understand how you calculated your « approximate change in stellar radiation ». Where does this come from?

- I102-103

Could you connect your results with the observational results on Mars obtained by Heavens et al. 2018 (<https://ui.adsabs.harvard.edu/abs/2018NatAs...2..126H/abstract>)?

- I104-106 : « In contrast to the effect on the outer edge, both our TL and nTL simulations result in a reduction in stratospheric water-vapour when including dust, demonstrating that the inward movement of the inner edge of the habitable zone is a ubiquitous feature of atmospheric dust »

Are you sure this mechanism holds when you are much closer (in insolation) to the inner edge of the HZ. On relatively dry (and thus dusty) planets, you would expect the water to migrate on the nightside of the planet. See Abe et al. 2011, Turbet et al. 2016, Kodama et al. 2018 (<https://ui.adsabs.harvard.edu/abs/2018JGRE..123..559K/abstract>), which makes your scenario TL continents more relevant. If you have a dusty atmosphere, you expect to decrease the stratospheric water vapour content (Fig 2b, red curve). But you also expect to warm the nightside of the planet (Fig 2a, red curve), which should boost the hydrological cycle on the nightside of the planet : more evaporation ; but also more efficient water flow on the surface (this can be seen in your Figure 3a and 3b where adding dust increases the surface temperature of the two nightside gyres which would drastically increase water ice flow from nightside to dayside), which should increase the area of wet lands, which could push back the inner edge of the HZ according to Kodama et al. 2018.

I understand your numerical results, but again I don't agree with your extrapolations on the position of the inner limit of the HZ.

- I-109-117 « In summary radiatively active atmospheric dust increases the size of the habitable zone for tidally-locked planets, both by moving the inner edge inwards and outer edge outwards »
...

I don't agree with this statement, because I think (cf my previous comments) that your results cannot simply be extrapolated to the conditions for which the HZ limits have been calculated.

- Figure 4 : Could you please specify the instrument used in these synthetic spectra? Is it NIRSPEC/Prism?

- I-266 – Methods

Did you consider lifted dust particles as a possible source of CCN?

Did you consider (or plan to consider, for a future study) the effect of sea salt particles that would be relevant even in the aquaplanet case?

Eventually, could you say a few words about the dust optical properties? What database did you use? And did you use the same database in the GCM and the PandExo simulations?

- I-282 : « The surface is essentially flat »

What do you mean by « essentially flat »?

- SUPPLEMENTARY MATERIALS

-I28-32 : « On a tidally-locked planet, the increase in temperature due to long-wave emission from the dust is stronger than the decrease in direct heat-transport from the day-side, therefore the night-side warms. On a non-tidally-locked planet, the decrease in heat-transport from equatorial regions is stronger than the increase due to long-wave emission from dust, therefore the polar regions cool. »

Again, you demonstrate that for an Earth-like atmosphere. But is this still true for a thick CO₂-dominated atmosphere?

-I39-42 : « As discussed in Lewis et al. 1 , land positioned on the day-side of a tidally-locked planet is the most interesting and useful case to consider, because any land located on the night-side is likely to be locked away under permanent ice-caps and so be unable to influence the planetary climate or carbon-silicate cycle »

I don't agree with the fact that any land located on the nightside is unable to « influence the planetary climate ». Such land can for example become a cold trap on which water or other condensable can accumulate, thus affecting the climate at the global scale.

See Menou 2013 (<https://ui.adsabs.harvard.edu/abs/2013ApJ...774...51M/abstract>)
Leconte et al. 2013 (<https://ui.adsabs.harvard.edu/abs/2013A%26A...554A..69L/abstract>)
or Yang et al. 2014 (<https://ui.adsabs.harvard.edu/abs/2014ApJ...796L..22Y/abstract>)

- I55-56 : « stratospheric water-vapour contents are quite high (similar to aquaplanet experiments 2) »

From your Figure S2, it seems that it reaches ~ 0.5ppm maximum, which is more than one order of magnitude lower than present-day Earth stratosphere which is considered to be pretty dry. Why do you consider your stratospheric water-vapour contents « quite high »??

- I58-59 : « The dust reduces the stratospheric water-vapour content down to amounts comparable to the 100% land simulations »

What does it mean? Where does the water come from in your 100% land simulations? Shouldn't be the water content equal to 0?!

- I72-74 : « Figure S4 presents the same results shown in Figure 4 but with the star moved further away, such that its apparent magnitude is similar to TRAPPIST-1 rather than Proxima Centauri. These simulated observations could therefore represent those of TRAPPIST-1e. »

Did you also adapt the size and spectrum of the star to TRAPPIST-1? Did you take the planetary parameters of TRAPPIST-1e? (in both the climate calculations and PandExo?)

How do the SNR of the CO₂ and other features compare to what have been calculated in the literature, e.g. in Lustig-Yaeger et al. 2019 (<https://ui.adsabs.harvard.edu/abs/2019AJ....158...27L/abstract>) and Fauchez et al. 2019?

Reviewer #3 (Remarks to the Author):

By incorporating airborne mineral dusts in general circulation models, the authors show that the extent of habitable zone can be substantially widened and the grey opacity of dusts tend to mute transmission features of biomarker molecules. Speaking as someone from the exoplanet community, the type of "dusts" that is much more detrimental to the detection of biomarker is the high-altitude clouds/haze primarily produced photochemically produced by UV flux. This is entirely different from the mineral dusts elevated from the planet's surface that the authors focused on in this work. The famous "detached" haze layer (400km above the surface) on Titan is believed to be hazes produced photochemically rather than mineral dusts lofted from the surface. It is this haze layer that gives rise to many interesting features of Titan such as the fuzzy appearance, orange coloration and strong forward-scattering. More importantly, in the near future biomarker molecules in exoplanets will be studied by transmission spectroscopy which probes the upper atmosphere. It is again the high-altitude hazes/clouds that most severely mute transmission features of biomarkers. The creation, growth and sedimentation of hazes and their effects on transmission spectroscopy have been extensively studied both in labs and in simulations (<https://arxiv.org/abs/1712.02808>, <https://arxiv.org/abs/1805.10488>, <https://arxiv.org/abs/1902.10151>)

The authors showed interesting results on the extent of habitable zone when mineral dusts is considered. I would be curious to see how high-altitude photochemical haze in GCM further affects the habitability. I understand that this would be an arduous undertaking. However, in its current form, I am afraid that the manuscript is not original enough or comprehensive enough that it warrants the publication in a high profile journal like NatureComms. I suggest that the authors submit the manuscript to Astrophysical Journal, Astrobiology or Icarus.

Reviewer #1 (Remarks to the Author):

This paper examines the climatic effect of dust loading into an atmosphere in two different scenarios. One with an Earth like world with increasing amounts of land to sea while keeping Earth's modern rotation rate. In the other case they look at a tidally locked Proxima Centauri b type planet with increasing amounts of land at the substellar point and "tiled."

This is a unique approach I've not seen before in the literature and has serious implications for observing terrestrial planets with JWST and other future instruments. I recommend publication, but I would like the authors to make a few changes. I don't need to see the manuscript again if they agree to these modest changes. Of course if other referees demand substantial revisions then it may be necessary to see it again afterward.

We thank the reviewer for their detailed assessment of our work, and have responded to all of the suggestions below as requested. Line numbers refer to the diff files.

1.) It is never discussed what type of ocean is used. Is it a simple mixed layer depth X ocean w/o horizontal heat transport? Is it a fully-coupled bathtub ocean? What is it?! Ocean dynamics are important in climate studies, and we should know what it being used.

A description has been added to the methods section (L344-346) – it is a slab ocean with no horizontal heat transport as with previous papers using the UM.

2.) Line 8: it is not until line 109 that the term "radiatively active mineral dust" is mentioned. I think that belongs here in the Abstract.

"radiatively active" has been added to the abstract as suggested.

3.) Lines 23-25 "But most studies so far have focussed on oceanic\aquaplanet" scenarios, because of the importance of the hydrological cycle and the definition of habitability requiring stable surface liquid water."

Yet there have been a few large parameter studies that have used non-aquaplanet scenarios, in particular: Yang et al. 2014 ApJL 787, L2, and Way et al. 2018 ApJS 239, 24. You might mention them and/or others.

We have included those after "Some studies have attempted to simulate the effects of the presence of land".

4.) Lines 31-32 "More specific treatments of land surface features such as topography remain unexplored."

This is a rather strong statement. It has been explored in a few areas, just not extensively. Your reference 7 (Del Genio et al.) does have Prox b simulations with an aquaplanet set up AND Earth topography (See their Table 4 and in particular #18 Day-Land) At least one more I am aware of is Way et al. 2016 GRL 43, 8376 where they explicitly compare a Venus topography to an Earth topography each

with an ocean of similar depth (see Simulation A versus C in Table 1).
Although there is no discussion of dust in either of these works.

We have revised the sentence to “More specific treatments of land surface features such as topography have only been briefly explored” and included citations to those works.

5.) I really like your Figure 1!

Thanks!

6.) From Figure 2 it looks to me like you did 4 simulations "TL tiled", 4 "nTL Tiled" and 5 "TL Continents." It would be nice to have a table of all of the simulations with their respective parameters listed. If you don't have room in the main text perhaps in the Suppl Mat?

This is a good idea – we have added Table 3 to the methods section which summarises the 32 experiments which are used in the paper.

7.) Lines 100-103: Sorry, but I don't quite understand this statement. First aren't you extrapolating to get the 70% simulation for "TL Continents?" Please say so if that's the case. Then you say 20-50% reduction in 2b. But if you mean only at 70% land fraction I don't see the same range. Could you be a bit more explicit? The same applies for the comment about 20-50 W/m².

To get the 70% value for TL Continents, we are interpolating between the 60% value (an 80% reduction) and 100% value (a 20% reduction) as shown on Fig 2b. The 20-50% reduction was then meant to encapsulate where all 3 experiments are at this land fraction, although you are correct that reading direct from the graph this is more like 30-60%. As the numbers are meant to be approximate, we have updated the text to match the graph.

We have tried to improve the description of how we infer the reduction in stellar flux from this (L112-116). We have done additional experiments (now listed in Table 3) with different stellar fluxes, to infer the logarithmic relation between stellar flux and stratospheric water vapour, i.e. actual reductions in stellar flux of 30-60W/m² lead to 30-60% reductions in stratospheric water vapour. Therefore, we infer that this reduction in stratospheric water vapour caused by dust is approximately equivalent to these changes in stellar flux.

We acknowledge that we are implying this relationship obtained at 70% land fraction holds for all land fractions. We wanted to quantify the change due to dust in terms of something more familiar to those interested in aspects of habitability, i.e. the change in stellar constant.

8.) Figure 3 caption: 8.3 km. Can you tell the reader what this is in pressure? hPa or mb would be fine.

The pressure is approximately 300hPa, and we have added this to the caption.

9.) Figure 4 b & d. Could you please add to the title of each 'plus biomarker gases' or something like that to make it easier for the reader to distinguish w/o having to read your caption so carefully?

We have added “plus biomarker gases” to the titles as suggested.

10.) Reference 24: the last author's name is missing a 'c.' Stamenkovic

We have fixed this typo.

11.) In your methods section regarding your dust parameterization. Nowhere is it explicitly mentioned whether you are using the default Earth parameterization for your dust, or if you used your model in interactive mode with with your dust sizes to generate a proper parameterization w/o Earth vegetation included. Please tell the reader.

We have now explicitly mentioned that we use the default Earth setup (L343). We think you're asking if we re-ran an Earth simulation without vegetation to retune the dust model(?), which we did not. There is nothing implicit in the dust parametrization about the presence or absence of vegetation – this is included via suppression of the dust uplift on surface types which contain vegetation, but because we do not use any of these we simply get the standard uplift from a bare-soil surface. Since surfaces such as bare soil are the most dominant source of dust on Earth (e.g. the Sahara), we believe the parametrization should be well-tuned against this as the primary uplift source. We have noted in the text that the parametrization naturally adapts to the absence of vegetation (L343).

12.) Table 2: Could you possibly add a column that lists the partial pressures for CO₂, O₂, O₃, CH₄ and N₂O in ppmv? It's easier to compare with other work w/o the reader having to do the work themselves. Also nowhere in the text does the reader learn what "MMR" means.

We have added the ppm values in addition to the mass mixing ratio, and also updated the caption to explain mmr and ppm.

13.) Line 282: Could you please be a bit more explicit about "The surface is essentially flat" so that the reader doesn't have to read Lewis et al. and hope it's the same? Some mention of roughness length would also seem appropriate since it will affect dust lifting.

We have rephrased this to “the land is at sea-level altitude, with zero orography and a roughness length of 1e-3 m for momentum and 2e-5 m for heat (although these are reduced when snow is present on the ground)”.

On to the Supplementary Material:

1.) Line 67 add TL: "... between TL land surface configurations..."

This has been corrected.

2.) Figure S4 certainly makes for depressing, but useful reading. Although it was only recently published it would be nice if you mentioned it in the context of Figure 5 from Fauchez et al. 2019 ApJ 887, 194, even if theirs is for a noise free spectrum, they do mention in their Table 4 how many transits you need for a given S/N for a specific feature.

Between your dust and their clouds and hazes, the prospects look bleak for detecting anything other than CO₂!

This is a nice suggestion – we have included a comment and reference to that paper (L100-102), which we think are in broad agreement with what we show.

Reviewer #2 (Remarks to the Author):

Overall, I think that the paper is well written, well organized and deals with a topic that has been largely overlooked in the literature. I am sure the results of this paper will be of interest to the community, as it brings first insights on the effects (and underlying mechanisms) that dust may have on the climate of temperate, terrestrial exoplanets.

We thank the reviewer for their thoughtful review of our work. We have addressed and responded to all of the comments, and we think that the resulting clarification of the major points below has significantly improved the work. Line numbers refer to the diff files.

However, there are two important points I would like the authors to clarify/correct/expand in their manuscript before I recommend it for publication.

My first main concern is about the numerical parameterization of the dust cycle. The authors mentioned in their manuscript that they used the mineral dust parameterization of Woodward et al. 2011 (technical report). The dust emission scheme seems to be tuned and designed with empirical coefficients to reproduce the Earth dust cycle, and I am concerned whether it can be applied as is on other planets.

It is well known in the planet Mars literature that the dust cycle (and in particular the dust injection) is difficult to model (Newman et al. 2002, Kahre et al. 2006, Mulholland et al. 2015, Bertrand et al. 2019) with 3-D GCMs because it depends on subtle sub-grid scale parameterizations.

Can the authors reproduce roughly the dust cycle of Mars with their model? (any reference of previous work done with the UK Met Office GCM? Any additional run?)

Can the authors explore how sensitive are their results to their dust emission parameterization? (e.g. by doing additional runs with various wind threshold friction velocities?)

Last, the authors assumed that the reservoir of dust is infinite. Is it reasonable especially for planets with low land fraction? A discussion on sources versus sinks would be welcome.

We have investigated the sensitivity of our results to various parameters within the scheme as suggested, and now include a discussion of these sensitivities in the main text (L140-146) and supplementary material (L14-22). The results are qualitatively unchanged, therefore we feel this shows that the basic message of the paper: mineral dust is important for habitability, is unchanged. Figure 1 below shows the results of two additional runs to support this. The first (panel c) shows a 10% reduction in a key tuning parameter, which links the point-source friction velocity used in the uplift equation to the grid-box mean friction velocity predicted by the model. Because of the nonlinearity of uplift to friction velocity, this tuning represents a ~30% reduction in uplift, and based on recent model versions can be considered a large but not unrealistic tuning of the scheme. As shown, it leads to a slight reduction in the effects we discuss (slight warming of the day-side and cooling of the night-side), but the change is small and the overall effect of dust is still a large perturbation to the NoDust simulation. The second (panel d) shows the same tuning to the uplift, but now with ~50% increased dust placed in the lowest 3 size-bins (and commensurate reductions in the upper 3 bins). This pushes the simulation back towards the control (panel b), as the reduced uplift is offset by the uplift being of the more radiatively active sizes. Again, the difference is small relative to the Dust vs NoDust comparison, but highlights that the actual distribution of available dust is a similar size uncertainty to the tuning of the parametrization itself.

Figure 1: Surface temperature from TL experiments with 100% land cover and (a) NoDust, (b) Control Dust, (c) reduced dust uplift, and (d) reduced uplift plus increased small dust. Also shown are the mean (solid) and maximum (dashed) 273 K contours.

There is no run of the UM or this dust scheme for Mars available. To do this would be a significant undertaking well beyond the scope of this paper. Additionally, the key challenge of parameterizing dust transport in the Martian atmosphere is to represent the interannual variability of dust – something GCMs still struggle to do, but not a key requirement for this work as we are focussed on climatological mean fields. Martian models have been developed and tuned over many years in the same way Earth models have – there is similarly no guarantee a well-tuned Martian model could simulate the dust cycle of Earth accurately. Accordingly, our GCM, which has been tuned for Earth, should be applicable to the regimes we are focussed on in this paper, i.e. surface pressures and equivalent surface stresses similar to Earth. Therefore, based on this choice, we believe a dust scheme tuned for Earth is the most sensible starting point.

It is correct that we have assumed an infinite reservoir of dust, although we feel that this is not actually a requirement to achieve the results. All we require is that the supply of dust is enough to support that which is suspended within the atmosphere (maximum 2g/m², Fig S1), and that dust deposited on surfaces unfavourable for uplift can somehow be returned to surfaces favourable for uplift. Such a scenario is essentially governed by the hydrological cycle, as dust deposited on the night side of the TL planet will be returned to the day side at the same rate as water deposited here. We have noted this in the methods section for clarity (L346-351).

My second main concern is about the interpretation and implication of their results regarding the habitability of terrestrial exoplanets (see also my more detailed comments hereafter).

Although I am convinced by the results obtained by the authors for 1 bar N₂-dominated Earth-like planet, I am not sure that they extrapolate well – unlike what is stated by they authors, even in the title of this manuscript – to the habitability of planets in general.

The primary focus of the current paper is approximately “Earth-like” planets, i.e. 1 bar surface pressure and nitrogen dominated atmospheres. There are 2 reasons for this choice – firstly, it represents the most well-known and understood planetary configuration modelled, and secondly it is the only planet type known to be inhabited. Therefore, we feel this is the key planetary atmosphere under which to investigate questions of habitability. We now clarify in the text that this is the main focus of the paper (L58-60).

We agree that how the effects of dust interact with the plethora of other variables in a planetary atmosphere which can affect habitability is an important topic, and one that future work will investigate in full detail. We have added some discussion of the points raised below to the main text (L187-198).

The authors performed indeed 3D GCM experiments for tidally locked planets with a 1 bar Earth-like atmosphere for which they showed that the dust cycle tend to stabilize the climate. This stems from the fact that dust reflects efficiently light on the dayside of the planet, thus cooling the dayside temperature of irradiated planets which reduces the stratospheric water vapour content and delay the moist greenhouse ; but warms the nightside, thus preventing from CO₂ surface condensation, thus stabilizing the climate of weakly irradiated planets.

(1) On relatively dry (and thus dusty) planets, you would expect the water to migrate on the nightside of the planet. See Abe et al. 2011, Menou 2013, Leconte et al. 2013, Yang et al. 2014, Turbet et al. 2016, Kodama et al. 2018, 2019. From that perspective, the scenario TL continents is relevant. You show for a 1bar N₂-dominated Earth-like atmosphere that if you have a dusty atmosphere, you expect to decrease the stratospheric water vapour content (Fig 2b, red curve). But you also expect to warm the nightside of the planet (Fig 2a, red curve), which should boost the hydrological cycle on the nightside of the planet. This means more evaporation, but also a much more efficient water flow on the surface. This can be seen in your Figure 3a and 3b where adding dust increase the surface temperature of the two nightside gyres which would drastically increase water ice flow from nightside to dayside. This should increase in turn the area of wet regions, which is the key ingredient for determining the position of the inner edge of the HZ according to Abe 2011, Kodama et al. 2018, 2019.

In summary, dust here would decrease the habitability of terrestrial planets.

This is an interesting suggestion, but also an effect we feel is unclear. For example, Haqq-Misra et al (2018, ApJ, 852, 67) Fig 1 shows that day-night temperature contrasts even on TL planets are very low near the inner edge of the habitable zone. Therefore, it is unlikely that planets very near the inner edge of the habitable zone will have large reservoirs of ice on the night-side that could be liberated by the presence of dust. In fact, we think this suggested mechanism is far more likely to occur near the outer edge of the habitable zone, widening it yet further.

We do agree that in the bi-stable state described by Leconte et al (2013), where large amounts of water are captured in cold traps (an effect in itself which widens the habitable zone), the presence of dust could offset this widening. However, it is still unclear whether this effect would be greater

than the dayside cooling limiting stratospheric water vapour. We thank the reviewer for mentioning the potential effects of ice caps, which we had not considered, and have included this in our discussion (L187-191).

(2) As the CO₂ partial pressure increases (e.g. up to several bars) in a terrestrial planet, the heat redistribution between day and nightside also increases (see for example Turbet et al. 2018 Fig. 11, Fauchez et al. 2019 Fig. 10). This should partly move a planet toward a 'nTL'-like configuration where the effect of dust on the nightside should be significantly lowered (see your Figure 2a, green curve).

For thick CO₂-dominated planets representative of planetary atmospheres used to estimate the position of the outer edge of the Habitable Zone (Kopparapu et al. 2013, Kadoya & Tajika 2019 and many other references), your Figure 2a seems to predict the opposite effect of your claim. In summary, dust here would again decrease the habitability of terrestrial planets.

The reviewer is correct that the effect of high pressures should be discussed, and we have added discussion on this to the text (L192-198).

The case of high CO₂ is certainly interesting, and we thank the reviewer for raising this point. We would note that such a scenario may not be possible on a tidally locked planet, since increasing CO₂ levels would further warm the day-side, increasing weathering activity and preventing further CO₂ build up in the atmosphere (e.g. Kite et al, 2011, ApJ, 743, 41).

We note that it is not just the day-night temperature contrast which is important for the behaviour of the TL planet; in other words reducing the contrast would not necessarily push the results towards those of nTL. The key difference between the configurations arises because of the dust uplift and transport to dark regions of the planet, both of which are much stronger in TL. We have clarified this in L75-89. Such a scenario would also happen with higher CO₂ partial pressure. Even with high CO₂ levels, the strong uplift will be maintained, and if anything, it would be enhanced as surface stress increases with pressure. Reduced day-night temperature contrasts may reduce the efficiency with which the super-rotating jet can distribute dust to the night side, but the redistribution should still happen and thus the infra-red warming of the night-side will still be present.

To end on that second main comment, new additional runs (e.g. taking self-consistently into account the land fraction as a function of the temperature of the nightside ; e.g. with thick multibars CO₂-dominated atmospheres) would be needed to support the claim that dust do really increase the habitability of terrestrial planets as stated in the title, in the abstract and throughout the text.

As noted above, we have clarified the scope of the current paper (L58-60), and with thanks to the reviewer have extended our description of the key mechanisms (L187-198). Examining detailed changes in many planetary parameters such as surface pressure, or even gravity, size, orbital configuration, for example, is beyond the scope of the present work. We have examined the sensitivity of our results to a very significant planetary configuration: land fraction/distribution, and show that for many cases dust is an important new effect hitherto ignored by terrestrial exoplanet climate models.

Last but not least, I have appended below a detailed list of remarks which mainly consists of minor remarks and additional clarifications on my two main concerns.

-l18 - « the first potentially-habitable terrestrial exoplanets1 »

The reference points to Wordsworth et al. (2011) which studied GJ581d. However, the planet was shown later not to exist. It was likely an artefact of stellar activity. See Robertson et al. 2014 (<https://ui.adsabs.harvard.edu/abs/2014Sci...345..440R/abstract>). It may be more relevant to cite the first non-disputed potentially habitable terrestrial-size planet : Kepler-62f. Ref : Borucki et al. 2013 (<https://ui.adsabs.harvard.edu/abs/2013Sci...340..587B/abstract>)

Many thanks for pointing this out – we have corrected the reference accordingly.

-l21 – ref 2 and 3

It may be more relevant to cite :

Dole 1964 (<https://ui.adsabs.harvard.edu/abs/1964hpfm.book.....D/citations>)

Kasting et al. 1993 (<https://ui.adsabs.harvard.edu/abs/1993Icar..101..108K/abstract>)

Barnes 2017 (<https://ui.adsabs.harvard.edu/abs/2017CeMDA.129..509B/abstract>)

This is a nice suggestion, and we have changed the references accordingly.

-l23-25 - « But most studies so far have focussed on oceanic “aquaplanet” scenarios, because of the importance of the hydrological cycle and the definition of habitability requiring stable surface liquid water. »

This is also because water-rich planets are one of the likely outcome of planetary formation models. See for instance Tian & Ida 2015 (<https://ui.adsabs.harvard.edu/abs/2015NatGe...8..177T/abstract>).

This is a good point, we have stated that and included the reference.

-l26-28 - « For a planet’s climate to be stable enough for a sufficiently long period of time to allow the development of complex organisms (e.g. around 3 billion years for Earth 9), the presence of significant land cover is likely required »The surface is essentially fla

This is extremely speculative! I suggest to rephrase « is likely required » by « may be required ».

This has been changed as suggested.

- l28 - « responsible for the stabilisation of CO2 levels »

Please add « responsible on Earth for the long-term stabilisation of CO2 levels »

This has been updated as suggested.

- l30 - « to simulate the effects of the presence of land 11,12,13 »

You may also cite the previously mentioned Turbet et al. 2016 (land planet) and Del Genio (ocean with continents) et al. 2019.

We have included these at the end of the sentence, after Proxima B, to which they both refer.

- l40 - « the presence of dust in the atmosphere of super-Earth planets has been reported 17 »

The reference 17 looks weird to me. The flat spectrum reported in Kreidberg et al. 2014 could be as well due to the presence of non-mineral clouds or photochemical hazes. Please give additional evidence/reference proving this must be dust.

We agree, and we have removed this statement.

- l85-86 - « The effect of dust is to sustain a CO₂ greenhouse effect at a lower stellar irradiance than when dust is absent »

Do you mean « to sustain a greenhouse effect »? Is this a typo?

We meant that by warming the night-side and thus preventing CO₂ condensation, the dust enabled the CO₂ greenhouse effect to continue to act at lower stellar irradiances. But we agree this could equally be seen as dust having its own greenhouse effect, so have altered the text as suggested.

- l86-87 : « implying that dust moves the outer edge of the habitable zone away from a parent star »

I don't agree with that. As the CO₂ partial pressure increases, the heat redistribution between day and nightside also increases. This can be observed in Turbet et al. 2018 (<https://ui.adsabs.harvard.edu/abs/2018A%26A...612A..86T/abstract>) Figure 11, Faucher et al. 2019 (<https://ui.adsabs.harvard.edu/abs/2019ApJ...887..194F/abstract>) Figure 10, and many other references.

So I guess this should partly move the planet toward a 'nTL'-like configuration where the effect of dust on the nightside should be significantly lowered (see your Figure 2a, green curve).

This could be tested by running your GCM with an atmosphere of e.g. 10bar dominated by CO₂.

As discussed above, we are referring here to the habitability of our investigated planets, and discuss the more general definition of the habitable zone later.

Also by looking at the Figure 17 of Forget et al. 2013 (<https://ui.adsabs.harvard.edu/abs/2013Icar..222...81F/abstract>) that did similar simulations for the case of early Mars, it seems the effect of dust is relatively low. Could you comment on that?

The ~10K change in mean surface temperature shown by Forget et al is quantitatively comparable to what we see in our nTL runs, and actually much greater than what we see in our TL runs, where mean temperatures only fall by <5K when dust is included. So whilst it may have appeared low in the context of the other variables explored by Forget et al, it was actually quite significant.

The key result for TL planets is that while the mean surface temperature is largely unaffected by dust, temperature contrasts are significantly reduced.

- Figure 2

I don't understand how you calculated your « approximate change in stellar radiation ». Where does this come from?

We have changed the description of how we infer the reduction in stellar flux (L112-116). We have done additional experiments (now listed in Table 3) with different stellar fluxes, to infer the logarithmic relation between stellar flux and stratospheric water vapour, i.e. actual reductions in stellar flux of 30-60W/m² lead to 30-60% reductions in stratospheric water vapour. Therefore, we infer that this reduction in stratospheric water vapour caused by dust is approximately equivalent to these changes in stellar flux.

See also our reply to reviewer 1 above.

- l102-103

Could you connect your results with the observational results on Mars obtained by Heavens et al. 2018 (<https://ui.adsabs.harvard.edu/abs/2018NatAs...2..126H/abstract>)?

In the Heavens et al. paper, the presence of dust isn't necessarily the mechanism leading to hydrogen escape, but rather it is deep convection, i.e. the meteorology of the storm itself, which is a product of Mars' thin atmosphere and strongly forced circulation. Given we are dealing with a very different atmospheric paradigm, we do not think that the above results would be present on planets with a 1 bar atmosphere. Future work, which would investigate the effects of different atmospheric pressure, gravity etc, should certainly examine the effect of very deep convection in large storms.

- l104-106 : « In contrast to the effect on the outer edge, both our TL and nTL simulations result in a reduction in stratospheric water-vapour when including dust, demonstrating that the inward movement of the inner edge of the habitable zone is a ubiquitous feature of atmospheric dust »

Are you sure this mechanism holds when you are much closer (in insolation) to the inner edge of the HZ. On relatively dry (and thus dusty) planets, you would expect the water to migrate on the nightside of the planet. See Abe et al. 2011, Turbet et al. 2016, Kodama et al. 2018 (<https://ui.adsabs.harvard.edu/abs/2018JGRE...123..559K/abstract>), which makes your scenario TL continents more relevant. If you have a dusty atmosphere, you expect to decrease the stratospheric water vapour content (Fig 2b, red curve). But you also expect to warm the nightside of the planet (Fig 2a, red curve), which should boost the hydrological cycle on the nightside of the planet : more evaporation ; but also more efficient water flow on the surface (this can be seen in your Figure 3a and 3b where adding dust increases the surface temperature of the two nightside gyres which would drastically increase water ice flow from nightside to dayside), which should increase the area of wet lands, which could push back the inner edge of the HZ according to Kodama et al. 2018.

I understand your numerical results, but again I don't agree with your extrapolations on the position of the inner limit of the HZ.

As discussed in reply to the major comments, we have included discussion of caveats to our argument in the text (L187-191). Whilst heating of the night side may have some effect on the hydrological cycle, it will still be set against the reduction in day-side temperatures and direct weakening of the hydrological cycle that we show. At worst it would seem that the two effects would negate each other. Additionally, if wetland areas were increased, the amount of dust uplift would be reduced, naturally reducing the warming of the night side that was leading to their

generation.

- I-109-117 « In summary radiatively active atmospheric dust increases the size of the habitable zone for tidally-locked planets, both by moving the inner edge inwards and outer edge outwards » ...

I don't agree with this statement, because I think (cf my previous comments) that your results cannot simply be extrapolated to the conditions for which the HZ limits have been calculated.

We have clarified our statements regarding the effect of dust on the habitable zone (L122-131). Whilst it is true that any statements about the habitable zone will be approximate given the inherent uncertainties in the topic, our statements stand as plausible effects of mineral dust on the habitability of terrestrial exoplanets. We are certainly not trying to provide a detailed reassessment of the limits of the habitable zone as defined in Kasting and others – as mentioned, this will certainly require many more simulations.

- Figure 4 : Could you please specify the instrument used in these synthetic spectra? Is it NIRSPEC/Prism?

It is NIRSPEC (G140M, G235M and G395M modes), and we have included this information in the text.

- I-266 – Methods

Did you consider lifted dust particles as a possible source of CCN?

We have not considered dust as a CCN source in this study – the CCN number is spatially and temporally fixed in our simulations, although it would be possible in future studies to utilise the aerosol modelling capabilities of the climate model to investigate this.

Did you consider (or plan to consider, for a future study) the effect of sea salt particles that would be relevant even in the aquaplanet case?

Again, we had not considered interactive sea-salt in this study, but it would be possible in future thanks to the aerosol model we have coupled.

Eventually, could you say a few words about the dust optical properties? What database did you use? And did you use the same database in the GCM and the PandExo simulations?

The dust optical properties are calculated from Mie theory assuming spherical particles, and each size division is treated independently within the radiation scheme. We have added this to the text (L333-335), and also provided an additional reference (Woodward 2001, JGR, 106, 18155) which gives further details on the optical properties, including refractive indices. The GCM and PandExo simulations are self-consistent, because we calculate the transmission spectrum within the GCM using the same radiative transfer scheme (albeit with more bands).

- I-282 : « The surface is essentially flat »

What do you mean by « essentially flat »?

We have rephrased this to “the land is at sea-level altitude, with zero orography and a roughness length of 1e-3 m for momentum and 2e-5 m for heat (although these are reduced when snow is

present on the ground)”.

- SUPPLEMENTARY MATERIALS

-128-32 : « On a tidally-locked planet, the increase in temperature due to long-wave emission from the dust is stronger than the decrease in direct heat-transport from the day-side, therefore the night-side warms. On a non-tidally-locked planet, the decrease in heat-transport from equatorial regions is stronger than the increase due to long-wave emission from dust, therefore the polar regions cool. »

Again, you demonstrate that for an Earth-like atmosphere. But is this still true for a thick CO₂-dominated atmosphere?

As discussed above, we believe that the mechanisms should still hold for thick CO₂ atmospheres, although the quantitative results will certainly differ. We have added some discussed to the text (L43-46).

-139-42 : « As discussed in Lewis et al. 1 , land positioned on the day-side of a tidally-locked planet is the most interesting and useful case to consider, because any land located on the night-side is likely to be locked away under permanent ice-caps and so be unable to influence the planetary climate or carbon-silicate cycle »

I dont agree with the fact that any land located on the nightside is unable to « influence the planetary climate ». Such land can for example become a cold trap on which water or other condensable can accumulate, thus affecting the climate at the global scale.

See Menou 2013 (<https://ui.adsabs.harvard.edu/abs/2013ApJ...774...51M/abstract>)

Leconte et al. 2013 (<https://ui.adsabs.harvard.edu/abs/2013A%26A...554A..69L/abstract>)

or Yang et al. 2014 (<https://ui.adsabs.harvard.edu/abs/2014ApJ...796L..22Y/abstract>)

We agree, and have rephrased the sentence. The point we were trying to make was that the land would be locked away under the accumulated condensable, and thus not “open” to the atmosphere, for either carbon-silicate weathering or dust uplift.

- 155-56 : « stratospheric water-vapour contents are quite high (similar to aquaplanet experiments 2) »

From your Figure S2, it seems that it reaches ~ 0.5ppm maximum, which is more than one order of magnitude lower than present-day Earth stratosphere which is considered to be pretty dry. Why do you consider your stratospheric water-vapour contents « quite high »??

We have clarified the language. What we meant was that the stratospheric water vapour contents were comparable to an aquaplanet scenario, i.e. in nTL-70% the stratosphere has not been significantly dried due to the presence of land, whereas it is an order of magnitude drier (~0.05ppm) in the 100% land simulations. Everything is lower than present day Earth because we don't have methane oxidation, which is a significant source of stratospheric moisture (see Boutle et al 2017, Fig 5b).

- I58-59 : « The dust reduces the stratospheric water-vapour content down to amounts comparable to the 100% land simulations »

What does it mean? Where does the water come from in your 100% land simulations? Shouldn't be the water content equal to 0?!

Apologies, we have clarified this in the methods. Any land surface initially has its soil moisture set to saturation, i.e. we are making the land very wet at initialisation. This introduces a small (fixed) inventory of water to the 100% land simulations, supporting a very weak hydrological cycle.

- I72-74 : « Figure S4 presents the same results shown in Figure 4 but with the star moved further away, such that its apparent magnitude is similar to TRAPPIST-1 rather than Proxima Centauri. These simulated observations could therefore represent those of TRAPPIST-1e. »

Did you also adapt the size and spectrum of the star to TRAPPIST-1? Did you take the planetary parameters of TRAPPIST-1e? (in both the climate calculations and PandExo?)

No, planetary parameters are exactly the same in each simulation just with the apparent magnitude changed, to allow ease of comparison between the features in the spectra from Fig 4 and Fig S4. We have clarified the text to make it clear that we are not directly trying to simulate TRAPPIST-1e here (L89).

How do the SNR of the CO₂ and other features compare to what have been calculated in the literature, e.g. in Lustig-Yaeger et al. 2019

(<https://ui.adsabs.harvard.edu/abs/2019AJ....158...27L/abstract>) and Fauchez et al. 2019?

Our results appear consistent with those presented elsewhere in the literature, and we have added a comment and reference as such to the text (L100-102).

By incorporating airborne mineral dusts in general circulation models, the authors show that the extent of habitable zone can be substantially widened and the grey opacity of dusts tend to mute transmission features of biomarker molecules. Speaking as someone from the exoplanet community, the type of “dusts” that is much more detrimental to the detection of biomarker is the high-altitude clouds/haze primarily produced photochemically produced by UV flux. This is entirely different from the mineral dusts elevated from the planet’s surface that the authors focused on in this work. The famous “detached” haze layer (400km above the surface) on Titan is believed to be hazes produced photochemically rather than mineral dusts lofted from the surface. It is this haze layer that gives rise to many interesting features of Titan such as the fuzzy appearance, orange coloration and strong forward-scattering. More importantly, in the near future biomarker molecules in exoplanets will be studied by transmission spectroscopy which probes the upper atmosphere. It is again the high-altitude hazes/clouds that most severely mute transmission features of biomarkers. The creation, growth and sedimentation of hazes and their effects on transmission spectroscopy have been extensively studied both in labs and in simulations

(<https://arxiv.org/abs/1712.02808>,<https://arxiv.org/abs/1805.10488>,<https://arxiv.org/abs/1902.10151>)

The authors showed interesting results on the extent of habitable zone when mineral dusts is considered. I would be curious to see how high-altitude photochemical haze in GCM further affects the habitability. I understand that this would be an arduous undertaking. However, in its current form, I am afraid that the manuscript is not original enough or comprehensive enough that it warrants the publication in a high profile journal like NatureComms. I suggest that the authors submit the manuscript to Astrophysical Journal, Astrobiology or Icarus.

We do not disagree that high-altitude photochemical hazes are important and interesting, that is why they are actively being worked on by many research groups around the world. It must be emphasised that photochemical hazes are very different to mineral dust in source regions, production mechanisms, size distribution, transport and radiative effects; it would therefore be erroneous to conflate hazes with mineral dust. Without any particular basis to think that mineral dust and photochemical haze may coexist and interact, a paper attempting to deal with both simultaneously would seem premature and likely only lead to further confusion about their fundamentally different physical characteristics.

Mineral dust is an important atmospheric constituent in its own right, known to be of key importance on the three Earth-like solar-system planets within the habitable zone (Venus, Earth, Mars). It has not been studied at all in the exoplanet literature (to our knowledge), despite the aforementioned importance on the three planets we know the atmosphere of more than any others. Earth and Mars additionally do not currently have any photochemical haze, and it has also been suggested that photochemical hazes are unlikely to exist on Earth-like planets orbiting M-dwarfs (Arney et al 2017 ApJ 836 49). We therefore strongly feel that a standalone study on dust has a strong motivation, and the presented results regarding the effect on the habitable zone and potential observations are of sufficiently wide impact and high profile to be relevant to the readers of Nature Communications.

REVIEWERS' COMMENTS:

Reviewer #1 (Remarks to the Author):

I am completely satisfied with the authors responses to my original referee report. There is no further need for revisions from my point of view.

Reviewer #2 (Remarks to the Author):

*** « We have investigated the sensitivity of our results to various parameters within the scheme as suggested, and now include a discussion of these sensitivities in the main text (L140-146) and supplementary material (L14-22).....»

→ Why did you choose to vary a 'key tuning parameter' by 10% only? And not 50%? 90%? Also, what is this 'key tuning parameter'?

→ Can you explicitly state in the Supplementary Materials which sensitivity tests (i.e. which parameters have been changed, and by how much) have been carried out? I suggest to add a table summarizing the sensitivity experiments, as well as to add a Figure (e.g. Fig.1 in your rebuttal) showing explicitly the results of these experiments.

Reviewer #3 (Remarks to the Author):

I thank the authors for answering my queries. We would like to request the authors to make an explicit clarification, in both the title and the manuscript, of photochemical haze and the mineral dust they are studying.

I have no further comments.

Reviewer #2 (Remarks to the Author):

*** « We have investigated the sensitivity of our results to various parameters within the scheme as suggested, and now include a discussion of these sensitivities in the main text (L140-146) and supplementary material (L14-22).....»

→ Why did you choose to vary a 'key tuning parameter' by 10% only? And not 50%? 90%? Also, what is this 'key tuning parameter'?

The parameter we altered was the multiplicative factor by which the friction velocity is enhanced to account for the fact that the equation relating friction velocity to dust uplift is derived for point sources, yet the GCM friction velocity is a spatio-temporal mean over the model grid-box and timestep. In the default setup this parameter has value 2.2, and we reduced it to 2 (which we now realise is actually a 20% reduction given its theoretical minimum is 1), based on knowledge of previous model tuning exercises that this represents a large, but not unrealistic, perturbation to the parameter, i.e. making this change in an Earth climate simulation would still produce a credible dust cycle, whilst larger changes would produce unrealistic results. Larger reductions also imply the scheme is becoming increasingly more unphysical, as they push it closer to being representative of point-source uplift, which is not the case in a GCM.

→ Can you explicitly state in the Supplementary Materials which sensitivity tests (i.e. which parameters have been changed, and by how much) have been carried out? I suggest to add a table summarizing the sensitivity experiments, as well as to add a Figure (e.g. Fig.1 in your rebuttal) showing explicitly the results of these experiments.

We thank the reviewer for this suggestion – we have added the two new panels in Fig.1 of the rebuttal as a new Supplementary Figure 2, and included the additional runs in Table 3. We have then expanded the paragraph in the Supplementary material about sensitivity tests as follows:

“To investigate these uncertainties, Supplementary Figure 2 shows the surface temperature from two additional simulations, for comparison to Figure 3. In Suppl. Fig. 2a, the multiplicative factor (k_1), used to scale the grid-box mean friction velocity into a point-source friction velocity applicable for use in the uplift equation, is reduced to 2 from its control value of 2.2. This change represents a large, but not unrealistic, variation in this parameter based on recent model tuning exercises, and due to the nonlinearity of uplift actually represents a $\approx 30\%$ reduction in uplift. In Suppl. Fig. 2b, this change is combined with a $\approx 50\%$ increase in dust in the lowest 3 size-bins (and commensurate reductions in the upper 3 bins), thus allowing increased uplift of the most radiatively important dust sizes. As show, both modifications lead to small quantitative changes to the results presented, but the qualitative difference from the NoDust simulation remains unchanged.”

Reviewer #3 (Remarks to the Author):

I thank the authors for answering my queries. We would like to request the authors to make an explicit clarification, in both the title and the manuscript, of photochemical haze and the mineral dust they are studying.

We thank the reviewer for this suggestion – we have changed the title to explicitly refer to “Mineral dust” and provided a footnote at the first use of the term mineral dust stating:

“Mineral dust is class of atmospheric aerosol lifted from the planetary surface and comprising the carbon-silicate material which forms the planetary surface. It should not be conflated with other potential material suspended in a planetary atmosphere, such as condensable species (clouds) or photochemical haze.”